# Caudal and Thalamic Segregation in White Matter Brain Network Communities in Alzheimer's Disease Population

Frederick Xu
*Department of Bioengineering*
*University of Pennsylvania*
Philadelphia, PA, USA
fredxu@seas.upenn.edu

Duy Duong-Tran
*Department of Mathematics*
*United States Naval Academy*
Annapolis, MD, USA
duongtra@usna.edu

Yize Zhao
*Department of Biostatistics*
*Yale University*
New Haven, CT, USA
yize.zhao@yale.edu

Li Shen
*Department of Biostatistics, Epidemiology, and Informatics*
*University of Pennsylvania*
Philadelphia, PA, USA
li.shen@pennmedicine.upenn.edu

*Abstract*—Neuroimaging studies have demonstrated that Alzheimer's disease (AD) is closely related to changes in neuroanatomy in the form of damage to both grey matter and white matter. However, the exact nature of AD's relationship with white matter anatomical deterioration is not fully understood at a systemic level. To investigate this knowledge gap, we constructed structural brain networks from ADNI-GO/2 diffusion tensor imaging (DTI) images with brain regions of interest (ROIs) as nodes and white matter connections as edges weighted by fiber density. The cohort consists of healthy control (HC), mild cognitive impairment (MCI), and clinically diagnosed AD subjects. By optimizing consensus modularity of structural brain networks at a subpopulation level to investigate community structure throughout a range of resolution parameters ($\gamma$), we observed a split of the reward-based decision-making module in the AD group at $\gamma = 1.3$, thus finding a 7th consensus community in the AD consensus brain network partition that was not present in that of MCI or HC populations. Upon further investigation, we found that thalamic and caudal regions were involved in the increased segregation of AD brain networks. These regions are implicated in regulation of decision-making processes, and their segregation from other decision-making regions is a novel finding in white matter biomarker studies of AD. Our study presents novel evidence that AD may be a disconnection syndrome at the mesoscopic structural level, with potential new avenues of exploration into the role of the thalamus and caudate that may reveal neural correlates of cognitive deficits in clinically diagnosed AD.

*Index Terms*—Alzheimer's Disease, Cognitive Systems, Diffusion Tensor Imaging, Magnetic Resonance Imaging, Neuroscience, Neuroimaging, Thalamus

## I. INTRODUCTION

Alzheimer's disease (AD) is the most common form of dementia, an irreversible progressive disease characterized by memory loss followed by deterioration of cognitive function and memory recall. There are currently 5.5 million people afflicted with Alzheimer's in the United States and the number is estimated to increase to 15 million by the year 2050 [1].

AD has become one of the leading causes of death due to the lack of effective treatment, with the death toll of Alzheimer's disease rising by 89% from 2000 to 2014 [2]. With an increasingly aging population at risk of developing the disease, efforts to uncover the progression of Alzheimer's have become ever more important.

Neuroimaging is at the forefront of studying AD, allowing researchers to map the distribution of pathaological traits within the brain at structural and molecular levels [3]. Observed imaging phenotypes are compared across subject populations, including healthy control (HC), subjects with significant memory concern (SMC) [4], subjects in pre-clinical stages of mild cognitive impairment (MCI) [5], and patients clinically diagnosed with Alzheimer's Disease (AD). Traditional approaches using magnetoencephalography (MEG), functional magnetic resonance imaging (fMRI), diffusion tensor imaging (DTI), and positron emission tomography (PET) have accumulated evidence pointing towards Alzheimer's disease being a disconnection syndrome, characterized by a reduction of efferent and afferent connections between cortical and subcortical regions of interest (ROIs) in AD patients when compared to other subject groups [6], [7].

A contemporary approach to investigating connectivity in AD is through network neuroscience, where studies observe altered properties of brain networks across subject groups defined using ROIs as nodes and their corresponding pairwise relationships quantified from functional and structural imaging modalities as edges [8]–[18]. Properties of brain networks can be characterized using local and global measurements that are sensitive to connectivity disruptions and hypothesized to possess neurobiological relevance [19]. In cortical thickness structural connectivity networks derived from magnetic resonance imaging (MRI) data, it has been found that network measurements related to small-world properties of clustering

coefficient and path length are perturbed in the AD groups [20]. Lowered local efficiency associated with AD has been found in fractional anisotropy white matter networks derived from DTI [21]. Structural networks constructed using fiber count and fractional anisotropy from DTI have also demonstrated increased shortest path length and decreased global efficiency of AD brain networks, related to a reduction in connectivity [22]. However, the effects observed from these network measurements have been inconsistent between studies and imaging modalities, with difficulties in replication [23].

Another approach to investigating network connectivity properties is via the mesoscale study of modularity, where modules correspond to clusters of densely connected nodes that are strongly coupled within the community but weakly coupled externally [24]. Changes in connectivity will be reflected in the detected communities, as reduced intermodular connections may result in increased segregation, while reduced intramodular connections may cause community structure to degrade and result in increased integration. Modularity studies in relation to Alzheimer's disease have primarily used functional signals taken from resting-state functional MRI (RS-fMRI), electroencephalography (EEG), and magnetoencephalography (MEG). Resting-state functional networks (RSNs) across diagnostic groups in Alzheimer's disease have been observed to possess decreased segregation in the frontoparietal and default mode networks due to decreased prominence of functional connectivity patterns within those two systems [25]. When investigating networks constructed from MEG, it was found that the number of modules and number of intermodular connections were reduced in AD subjects [26]. Isolated studies observing modularity changes in structural networks in relation to AD have been sparse, as former research on the topic involved structural data statistically coupled to functional modularity but not studied directly [27] or structural modularity as a predictor in regression models [28].

For this study, we used fiber density as edge weights. We investigate changes to the number of modules and ROI community membership within consensus partitions of representative fiber density networks across subject populations within an Alzheimer's disease cohort. We expect to observe alterations in two properties of community structure across disease groups: 1) the number of communities detected, and 2) the ROI membership of detected communities. In particular, we expect to observe these changes in communities with ROIs implicated in memory recall and decision-making that support the hypothesis that AD is a disconnection syndrome at a structural level.

## II. MATERIALS AND METHODS

### Data and Preprocessing

We selected subjects that have available data in DTI scans, T1-weighted structural MRI (sMRI), and demographic data from the Alzheimer's disease neuroimaging initiative (ADNI-GO/2) database [29]. The subject population consisted of healthy control subjects (n = 76), MCI subjects (n = 68),

TABLE I
ADNI GO-2 COHORT (N=173) POPULATION DEMOGRAPHICS

| Variable | HC | MCI | AD | p |
|---|---|---|---|---|
| Avg. Age | $73.3 \pm 7.89$ | $72.5 \pm 7.40$ | $73.0 \pm 5.81$ | 0.805 |
| Avg. Edu. (Yr.) | $16.0 \pm 2.70$ | $15.6 \pm 2.85$ | $15.3 \pm 2.91$ | 0.939 |
| Num. M. | 42 | 37 | 20 | - |
| Num. F. | 34 | 31 | 9 | - |
| % Right-hand | 89.5% | 91.1% | 89.7% | - |

TABLE II
ADNI GO-2 COHORT (N=173) ETHNIC REPRESENTATION

| Group | Nat. Am. | Asian/Pac. Isl. | Afr. Am. | W. Cauc. | Multi |
|---|---|---|---|---|---|
| HC | 1 | 3 | 1 | 70 | 1 |
| MCI | 0 | 3 | 7 | 56 | 2 |
| AD | 0 | 0 | 2 | 27 | 0 |

and AD (n = 29) patients. The sex distribution was 99 male and 74 female subjects. The average age of the total population was 73.0 years with the AD population averaging at 72.5 years, the MCI population averaging at 72.0 years, and the HC population averaging at 73.3 years. We tested for group differences in continuous demographic variables using ANOVA (Table I). No significant difference in ages was detected across population groups ($p = 0.805$). Average number of years of education received was also consistent across population groups ($p = 0.939$), with HC averaging 16.0 years, MCI averaging 15.6 years, and AD averaging 15.3 years. Subjects were also primarily right-handed (HC = 89.5%, MCI = 91.1%, AD = 89.7%). Ethnic distribution in each subject group is predominantly white caucasian (Table II). The DTI data were entered into an image processing pipeline, including denoising, motion-correction, and distortion-correction using an overcomplete local principal components analysis (PCA) [30]. Probabilistic white matter fiber tractography was then performed using a streamline tractography algorithm called fiber assignment by continuous tracking (FACT) [31].

sMRI images were then registered to lower resolution b0 volume of the DTI data using the FLIRT toolbox in the FMRIB Software Library (FSL) [32] and 83 cortical and subcortical brain regions of interest (ROIs) were extracted based on the Lausanne 2008 Scale 33 Parcellation [33]. To define the network edges, we use fiber density as described by [34], [35]. Specifically, the number of the fibres (NOF) connecting each pair of ROIs $(i, j)$ and each ROI's surface area (SA) were obtained, and the fiber density in the connection was obtained by dividing NOF between ROIs $(i, j)$ by the average SA of regions $i$ and $j$ [34]. This normalization will correct for a trend observed where larger ROIs have relatively higher fiber counts. The final networks based on 83 ROIs were constructed using the fiber density of tracts connecting between pairs of ROIs. All obtained edges are retained, without an a priori thresholding procedure to avoid a known issue of graph theoretical measure sensitivity to edge threshold selection [36].

To study the changes in modularity across diagnostic populations, the 173 subjects with diagnostic labels of HC, SMC,

early MCI, late MCI, and AD were further stratified into three groups of HC, MCI, and AD to match more recent definitions of AD labeling from the ADNI [37]. The HC group consisted of 76 subjects, including HC subjects without symptoms (n=37) and SMC subjects (n=19) that were self-reported to have cognitive decline without objective cognitive impairment [4]. The MCI group consisted of early MCI (n=18) and late MCI (n=50) subjects. The final AD group consisted of 29 subjects clinically diagnosed with Alzheimer's disease.

*Optimizing Modularity of Averaged Networks*

We sought to find community structures within the representative networks of each subject population. Modularity was optimized using the community Louvain [38] implementation in the Python version of the Brain Connectivity Toolbox (BCT) [19] using a quality function containing a resolution parameter $\gamma$ [39]. Subject fiber density matrices within each population were bootstrapped, resampling 80% of the population with replacement, and averaging the density values to create mean networks. These bootstraps were conducted 100 times to generate an ensemble of partitions for each subject group. By creating an ensemble, we can account for the variability of optimal partition outputs from the Louvain algorithm [38]. Bootstrapping and averaging networks will also account for within-group variability of subjects. The resolution parameter $\gamma$ was varied from 0 to 5 at increments of 0.1 to capture a wide range of partition structures. For each gamma, the average final $Q$-value within each subject group, the average number of modules within each subject group, and the average pairwise partition distance between subject groups defined by normalized mutual information [40] were calculated. The implementation of partition distance can be found in the BCT package. Elbow transitions were then found in these metrics varying by $\gamma$ to obtain the optimal resolution parameter.

*Construction of Association Matrices*

The goal of the experiment is to obtain consensus community partitions for each of the subject groups that are statistically significant and capture group patterns. To do this, we first created association matrices ($\mathbf{A_p}$) for each subpopulation by counting the number of times a node $i$ shared a community assignment with another node $j$ normalized by the number of partitions ($n = 100$) [41]. Entries in the association matrix $\mathbf{A_p}(i,j)$ denote the probability that nodes $i,j$ would share a community for the population $p$. To ensure statistical significance, the averaged association matrices were then thresholded using a post-optimization null model [42]. The null model was obtained by performing random permutations of the 100 partitions obtained previously for each subpopulation, reassigning each node's community membership at random with uniform probability, from which the null-model association matrix ($\mathbf{A_{null,p}}$) was then computed. $\mathbf{A_p}$ was then thresholded, where values that were less than the maximum value expressed in the corresponding $\mathbf{A_{null,p}}$ were set to 0. This process was repeated for partitions obtained for each subject group to obtain statistically significant association matrices.

*Optimizing Consensus Partitions from Association Matrices*

To obtain consensus modules, community Louvain was run on the thresholded averaged association matrices. This method has been proven to provide stable consensus partitions robust to noise and heuristical variation in modularity optimization [41] [42]. The effect of resolution parameter was again explored, over the range of $\gamma_{\mathrm{con}} = 1$ to $\gamma_{\mathrm{con}} = 5$ in increments of 0.1. To select the optimal resolution parameter, average number of groups, average group size, and minimum group size were computed for each population. The goal was to select a $\gamma_{\mathrm{con}}$ that would yield 1) communities of reasonable size, 2) an inflection point where community structure shows significant change, and 3) a number of communities that may potentially represent the 7 RSNs in the Yeo Atlas [43] plus an additional subcortical ROI group. Consensus partitions were thus obtained for each subject group for further functional annotation.

*Functional Annotation of Modules*

A mapping of the 7-RSN Yeo Atlas [43] with an additional interhemispheric subcortical structure network (thalamus, caudate, putamen, pallidum, hippocampus, amygdala, accumbens area, ventral diencephalon) to the Lausanne Scale 33 Parcellation was used for functional annotation of modules. ROI members of the obtained consensus modules were then assigned to a Yeo Atlas system. Ranked modes were computed within each module, with the top 2 most represented RSNs assigned to each module.

## RESULTS

*Resolution Parameter Selection*

Optimal module identification has been demonstrated to possess a resolution limit, where it is impossible to detect modules of a certain size [44]. As such, we investigated the effect of tuning a resolution parameter $\gamma$ when optimizing a quality function [39] for averaged fiber density networks obtained from bootstraps (80% resampled) of subject networks in each diagnostic population. To find the optimal $\gamma$, we observed the average number of modules within the population, average $Q$-value within the population, and average partition distance between populations when varying $\gamma$ in increments of 0.1 from 0 to 5. It was found that the average $Q$-value decreased monotonically with a slight elbow transition at $\gamma = 1$, and the average number of modules increased linearly with $\gamma$, for all three populations of subject networks (Fig. 1a,b). We considered these results to be insufficient to inform optimal resolution parameter selection, therefore we studied a third measurement: the distance between partitions of two different subject groups (HC vs. MCI, HC vs. AD, MCI vs. AD) defined by normalized mutual information [40]. A monotonic relationship was found, where the partition distance between groups increased as $\gamma$ increased, with an elbow transition yielding a plateau beginning at $\gamma = 1$. In observance of this plateau, $\gamma = 1$ was chosen as the optimal resolution parameter, where the partition distance between populations was at near maximum while maintaining a reasonable number of modules

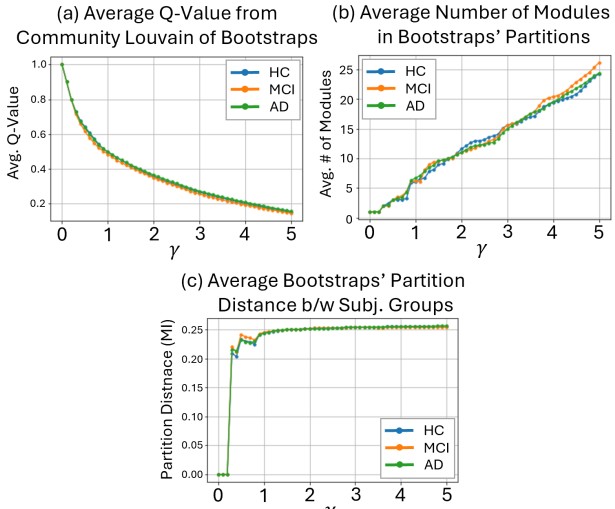

(a) Average Q-Value from Community Louvain of Bootstraps

(b) Average Number of Modules in Bootstraps' Partitions

(c) Average Bootstraps' Partition Distance b/w Subj. Groups

Fig. 1. When tuning the resolution parameter $\gamma$ for the bootstrapped network community detection procedure, we tracked the metrics of modularity quality ($Q$-value), number of modules within each subject population, and average partition distance between subject populations, and reported the average across the 100 bootstraps performed within each population. (A) We found that $Q$ decreased monotonically as $\gamma$ decreased, with an elbow-like transition at around $\gamma = 1$. (B) We found that the average number of modules increased monotonically with increases in $\gamma$. (C) The average partition distance defined with normalized mutual information between subject groups reached near-maximal values at $\gamma = 1$ before plateauing. This elbow transition indicates that $\gamma = 1$ would be the ideal selection of the resolution parameter, where the partition distance between groups is maximized while the number of communities can be reasonable.

within each population (Fig. 1c). It is noted that $\gamma = 1$ is often chosen as a default case, as it is equivalent to maximizing the standard quality function $Q$ [45] and represents a natural trade-off between small and large communities [24] [39].

*Consensus Partitions from Association Matrices*

To analyze differences in modularity among populations, we sought to find representative consensus modules for each subject group via modularity optimization of thresholded association matrices. The associations were gathered from module assignments with $\gamma = 1$ for 100 averaged networks constructed from bootstraps (80% resampled with replacement) of each diagnostic population (Fig. 2a), and were then thresholded against association matrices of post-optimization permutation null models (n = 100 random permutations, Fig. 2b). Using community Louvain and tuning a resolution parameter $\gamma_{\mathrm{con}}$ from 1 to 5 to find an optimal selection for the resolution parameter for consensus partitions, it was observed that the number of communities was largely stable from $\gamma_{\mathrm{con}} = 1$ to 3 for all three populations (Fig. 3a). A transition was observed at $\gamma_{\mathrm{con}} = 1.3$ where the AD consensus modules increased from 6 to 7. Beyond $\gamma_{\mathrm{con}} = 3$, the module assignments became unstable, with the average size of a community dropping rapidly (Fig. 3b). Upon further investigation, it was found that the minimum community size rapidly descended to 1 after $\gamma_{\mathrm{con}} = 3$, indicating instability in the community assignments at high resolution parameter values (Fig. 3c). Thus, we chose

to investigate the consensus modules found for each subject group at $\gamma_{\mathrm{con}} = 1$ and $\gamma_{\mathrm{con}} = 1.3$, where the community structures are the most stable and persistent across the resolution range.

*Community Assignment Changes in Consensus Partitions Across Resolutions*

Upon computing consensus modules, we sought to investigate the differences that existed across the subject groups. When optimizing modularity with $\gamma_{\mathrm{con}} = 1$ on the association matrices, 6 communities were discovered in each of the subject groups (Fig. 4a). The main differences observed between HC and MCI subject groups were an assignment switch of the brain stem from community 4 to community 1, and a switch of left supramarginal from community 5 to community 6. However, in MCI vs. AD, the left supramarginal gyrus returned to its HC community assignment. Between MCI and AD, the left transverse temporal gyrus saw an allegiance switch from community 6 to community 5. Lastly, MCI vs. AD saw assignment switches of left and right caudate along with left and right thalamus from community 4 to community 1.

At $\gamma_{\mathrm{con}} = 1.3$, we observed a 7th module in subjects with AD. The aforementioned differences described in MCI vs. AD in left and right (L/R) caudate and L/R thalamus were captured with their segregation into the 7th community (Fig. 4b). HC vs. MCI communities remained consistent between the two tested $\gamma_{\mathrm{con}}$ values. We thus note that the partitions obtained at a resolution of $\gamma_{\mathrm{con}} = 1.3$ uncover increased segregation in the representative AD brain network (Fig. 4e), and warrant further investigation with functional annotation to identify key differences in the AD consensus partition.

To investigate functional annotation of each module, we mapped each module's members to 7 functional RSNs based on the Yeo Atlas [43] with an additional subcortical group. Each module was labeled with the 2 most represented RSNs within the community. The segregation of L/R thalamus and L/R caudate in AD captured at $\gamma_{\mathrm{con}} = 1.3$ depicted these two subcortical regions separating from community 4, which consists mainly of Limbic and Subcortical system ROIs. Further investigation of community 4's members reveals that the regions are primarily involved with reward-based decision-making (L/R accumbens area, L/R medial orbitofrontal, L/R frontal pole, L/R rostral anterior cingulate) [47], and as such, we coin this community of ROIs as the reward-based decision-making module, which sees increased segregation in AD with the reassignment of the thalamic and caudal structures into a separate module.

## DISCUSSION

It has been hypothesized that Alzheimer's disease is a disconnection syndrome, where symptoms of memory recall deficits and cognitive dysfunction may be related to a loss of structural white matter connectivity in key systems. To investigate that notion, we sought to optimize modularity of fiber density networks across a range of resolution parameters and detail the observed changes in the number of communities

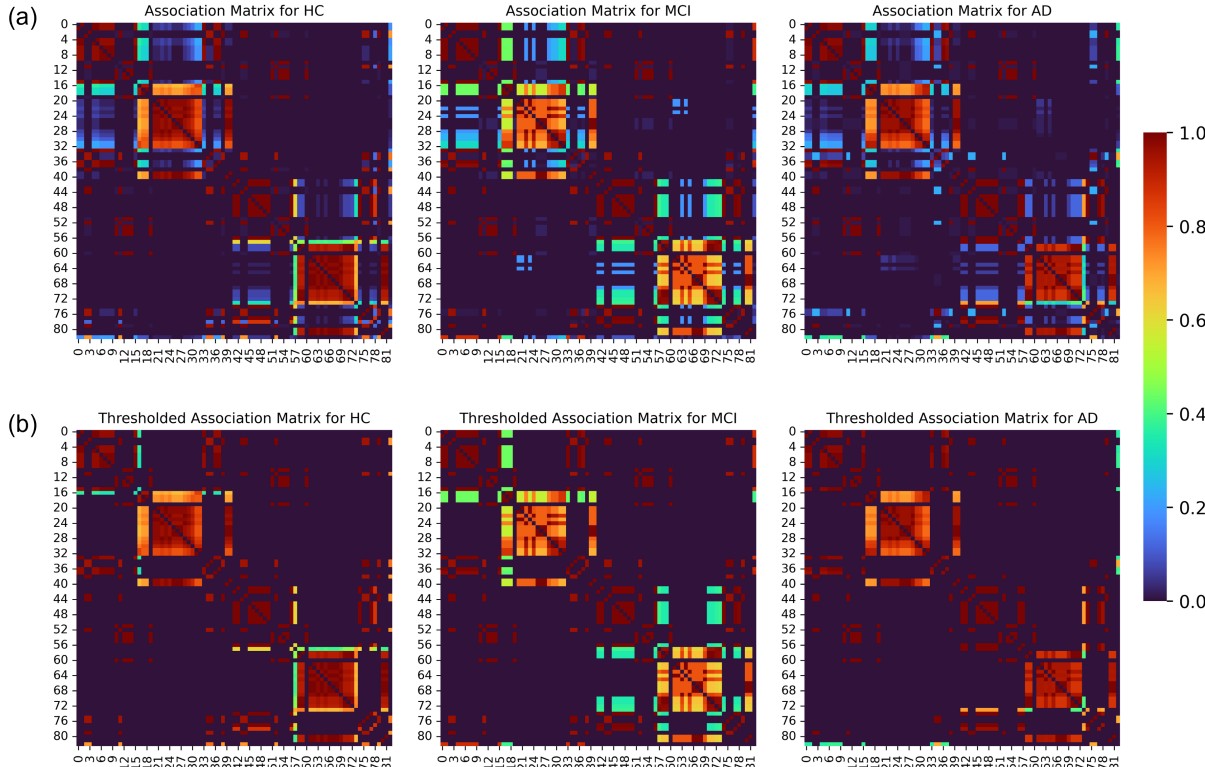

Fig. 2. Association matrices for each subject population were obtained, where the number of times a node shared a community assignment with another node was recorded. (a) In the top row, the original association matrices showed many low frequency co-assignments, with faint relationships between many of the regions. (b) After thresholding against post-optimization null models, we observed that many of the lower-association values were removed, resulting in only statistically significant associations post-thresholding.

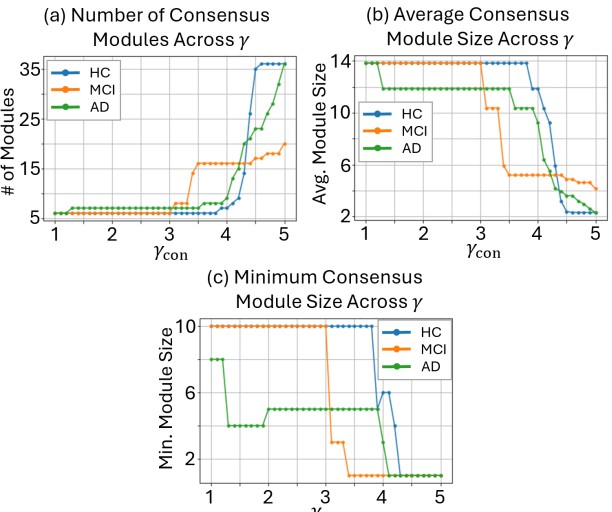

Fig. 3. For the consensus modules, we tracked the (a) number of modules, (b) the average module size in each partition, and (c) the minimum module size in each partition obtained throughout the $\gamma$ parameter tuning.

and membership of communities among HC, MCI, and AD groups. Key differences were found in the AD group when compared to HC and MCI. When investigating consensus partitions for each subject group, we found a largely consistent set of 6 modules across all subject groups, with a 7th stable module in AD at $\gamma_{con} = 1.3$. The source of the increased segregation in the AD group came from the separation of two subcortical structures, the thalamus and the caudate in both hemispheres, away from the reward-based decision-making module.

*Modularity Changes in Alzheimer's Disease*

Previous studies of modularity in brain networks in AD subjects have primarily investigated functional connectivity during resting state. A resting-state fMRI study showed that there was decreased segregation of frontoparietal and default mode networks [25]. In MEG, it was found that the parietal lobe became more segregated, with a reduction of intramodular connections in AD subjects [26]. It is to be noted that modularity of structural human brain networks has been demonstrated to change with age [48]. However, due to the controlled age across subject populations within the ADNI-GO/2 cohort ($p = 0.805$, Table I), the observed differences between diagnostic populations are unlikely to be due to age. Thus, the findings in our study of modularity in fiber density networks reveal a new phenomenon of the segregation of the thalamus and caudate related to AD. The notion of integration of modules has been associated with the performance of higher-order cognition and

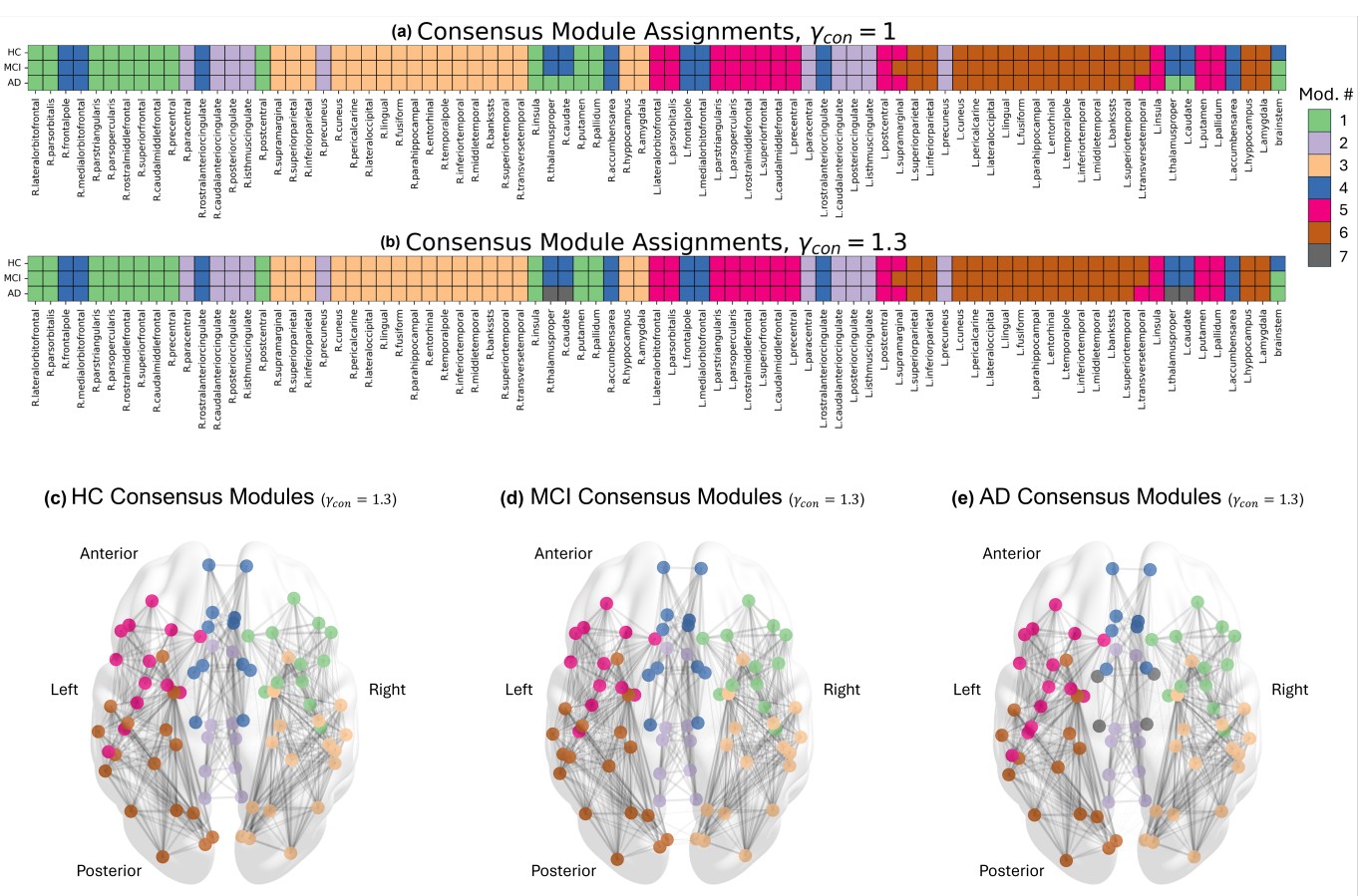

Fig. 4. Consensus module assignments were obtained and plotted using a color code, and differences among the populations were observed. (a) Six consensus modules were found for $\gamma_{con} = 1$ in all subject groups. We observed an assingment switch of left supramarginal gyrus and brainstem when comparing MCI to HC. AD saw L/R thalamus and L/R caudate switch allegiance to group 1 compared to MCI. (b) At $\gamma_{con} = 1.3$, 7 consensus modules were found in AD, while the same 6 modules persisted for HC and MCI. Here, the segregation of L./R. thalamus and L./R. caudate were observed. (c-e) A axial view of consensus brain networks and consensus modules were plotted in 3D space for each subject group using matplotlib [46] overlaid on top of FreeSurfer's averaged brain surface to provide a spatial depiction of the obtained communities.

working memory [49]; however, it is balanced with segregation to maintain specialization of function [50].

In the present study, we observed that the detected structural communities across all three subject groups are largely split between left and right hemispheres with a set of central brain modules connecting the two hemispheres (Fig. 4c-e), with a disruption to the interhemispheric relay regions in AD where the thalamus and caudate are segregated into their own community. Therefore, observing segregation in AD subjects not present in HC subjects may indicate an upset in the integration-segregation balance, providing a potential point of investigation.

*Separation of Key Decision-Making Structures*

It has been demonstrated in previous studies that the thalamus and caudate regions are implicated in the decision-making pathways. The thalamus has traditionally been considered a relay for neural connections; however, a recent study in mice showed that the mediodorsal portion of thalamus is also in close communication with the prefrontal cortex, regulating rule-based information involved in decision making [51] [52]

[53]. Recent reviews of structural changes in neuropsychiatric disorders have also corroborated evidence of mediodorsal thalamic disruption in schizophrenia subjects [54]. A study on electrode signals in monkeys during an AX continuous performance task, a commonly used cognitive deficit assessment for schizophrenia, revealed that mediodorsal thalamus neurons were implicated in response selection [55]. As such, it can be surmised that the thalamus is a key structure in mediating decision-making processes in addition to its function as a relay region, and its disruption could potentially be related to the symptoms observed in neurodegenerative disease. Accordingly, Alzheimer's disease is accompanied with a decline in decision-making performance, with decreased ability in AD subjects to make decisions under risk compared to MCI subjects [56]. Therefore, we speculate through our study that the disturbance to thalamic participation in the reward-based decision-making module observed in AD subjects compared to HC and MCI subjects may be related to cognitive deficits present in AD.

The second set of structures that showed segregation from

the reward-based decision-making module were the left and right caudate. The region has been found to assess outcomes of goal-directed behavior, as such it plays a significant role in mediating action schemas [57]. Given this knowledge, the observed isolation of the caudate away from the reward-based decision-making structures observed in AD subjects may represent another neural correlate of reduced decision-making abilities in AD patients.

*Limitations*

When performing modularity optimization, it is known that the optimal community assignment is a computationally intractable problem, with optimization algorithms often making use of a heuristic to achieve a partition of approximately maximal quality [24]. As such, the optimal community assignments may differ based on the approach used. However, in the present study, we have employed the association matrix to increase the stability of heuristic modularity optimization [42], thus significantly mitigating the variance that arises from community Louvain.

It must also be recognized that network analysis is sensitive to the choice of ROI parcellation scheme [58], network link density [59], and thresholding of edge strengths [36], procedures which are not fully standardized in the field. The exact nature of modularity's sensitivity to network construction protocol will be investigated in future work.

Lastly, the present study was conducted in one cohort of ADNI. While the AD cohort used is limited in size due to imaging data availability, the cohort is sufficiently large in comparison to those of previous works that make use of a modularity-based study paradigm [25] [26]. To address cross-cohort generalizability, we aim to investigate a validation cohort in future studies.

## CONCLUSION

Through the investigation of modularity of representative white matter fiber density networks, we have identified topological changes in subjects with clinically diagnosed Alzheimer's disease in a module associated with reward-based decision-making. The study has uncovered a potential role of the thalamus and caudate structures, which become segregated from the decision-making ROIs in AD subjects, that can be further explored to explain cognitive deficit symptoms observed in clinically diagnosed AD. Our results also confirm that Alzheimer's disease is indeed a disconnection syndrome at the structural level, with the novel exploration into the segregated community structure of fiber density networks at greater disease severity.

## ACKNOWLEDGMENT

This work was supported in part by the National Institutes of Health grants T32 AG076411, RF1 AG068191, U01 AG066833, U01 AG068057, and R01 AG071470. Data collection and sharing for this project was funded by the Alzheimer's Disease Neuroimaging Initiative (ADNI) (National Institutes of Health Grant U01 AG024904) and DOD ADNI (Department of Defense award number W81XWH-12-2-0012).

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
