# OpenReview forum: "Caudal and Thalamic Segregation in White Matter Brain Network Communities in Alzheimer's Disease Population"
_IEEE.org/EMBS/BHI/2024/Conference — IEEE BHI'24_

### Official Review · Reviewer_nsMo · 2024-08-05
**Review of submission 78**

**Overall Rating:** 7
**Confidence:** 4

**Other Quality Metrics:**

(a)Clarity of writing:4
(b) clinical significance: 4
(c) Methodological Novelty: 5
(dExperiments and Results: 4

**Questions For The Authors:**

See above.

**Strengths:**

1. Utilizes advanced neuroimaging techniques (DTI) to construct structural brain networks, providing a detailed exploration of white matter deterioration in AD.
2. Identifies a unique pattern of segregation involving the thalamus and caudate from the reward-based decision-making module in Alzheimer’s disease patients, which is not observed in other subpopulations like MCI and HC.
3. Provides evidence supporting Alzheimer's disease as a disconnection syndrome at a mesoscopic structural level, enriching the current understanding of the disease’s neuroanatomical impacts.

**Summary Of The Paper:**

Alzheimer's disease (AD), the most prevalent form of dementia, is an irreversible condition marked by progressive memory loss and cognitive decline, currently affecting 5.5 million people in the U.S., a number projected to triple by 2050. As a leading cause of death without effective treatments, understanding AD's progression through neuroimaging has become crucial. Techniques like MEG, fMRI, DTI, and PET highlight AD as a disconnection syndrome, characterized by diminished neural connections within the brain. Recent studies employing network neuroscience have analyzed altered brain network properties in AD, revealing disruptions in connectivity and network efficiency. This paper further explores these phenomena using network analysis on MRI data to study changes in community structures and connectivity within the brain, hypothesizing that AD-related structural changes support the disconnection syndrome hypothesis, particularly affecting regions linked to memory and decision-making.

**Weaknesses:**

1. Modularity optimization, crucial for the study's methodology, is computationally intractable, relying on heuristic approaches that only approximate the optimal community assignments, leading to potential variability in the results.
2. The results are highly sensitive to the specific methods used for network analysis, including the ROI parcellation scheme, network link density, and the thresholding of edge strengths.
3. Due to the sensitivity to various methodological factors, the modularity results may differ when different network construction schemes are applied, which can affect the reproducibility and reliability of the findings.
4. The reliance on heuristic methods for achieving community partitions suggests that the findings might not represent the true underlying biological structures or interactions within the brain, possibly limiting the accuracy of interpretations.
5. The study's conclusions are based solely on data from the Alzheimer’s Disease Neuroimaging Initiative (ADNI-GO/2) database. Expanding the research to include larger and more diverse datasets would enhance the generalizability and robustness of the findings, making the conclusions more universally applicable and convincing.

---

### Official Review · Reviewer_jpSA · 2024-08-09
**Review 78**

**Overall Rating:** 7
**Confidence:** 2

**Other Quality Metrics:**

(a) Clarity of writing: good
(b) Clinical Significance: good
(c) Methodological Novelty: good
(d) Experiments and Results: good

**Questions For The Authors:**

- I suggest to briefly describe the characteristics of the dataset used. Are the patients of the ADNI-GO/2 all included in the present analysis or are they selected in some way?

**Strengths:**

The study provides interesting insights into the pathophysiology of Alzheimer disease under the hypothesis of the disease as a disconnection syndrome. The paper is well written and the methodology used appears convincing even though some revisions are needed.

**Summary Of The Paper:**

The present paper investigated structural brain networks from diffusion tensor imaging (DTI) in patients with Alzheimer disease compared with healthy control subjects and subjects with mild cognitive impairment.

**Weaknesses:**

- The abstract needs revision. Indeed, acronyms should be defined. Moreover, some numerical results should be given.
- Duplicated information are reported in the first and in the last paragraph of the "Data and preprocessing" subsection.

---

### Official Review · Reviewer_CzPV · 2024-08-11
**This paper investigates the structural brain network changes in Alzheimer's disease (AD) by examining white matter connectivity using diffusion tensor imaging (DTI) data. The findings suggest that AD may be characterized as a disconnection syndrome at the mesoscopic structural level, contributing insights to the understanding of the disease.**

**Overall Rating:** 8
**Confidence:** 3

**Other Quality Metrics:**

Clarity of writing: Great
Clinical Significance: Good
Methodological Novelty: Great
Experiments and Results: Good

**Questions For The Authors:**

How might the findings change if a different ROI parcellation scheme or network construction method were used?

Given the limited sample size, especially in the AD group, how confident are you in the generalizability of these findings?

Are there any plans to extend this research to include longitudinal analyses?

How do the observed structural changes correlate with clinical measures of cognitive decline in AD?

**Strengths:**

Provides a new perspective on AD by identifying the segregation of the thalamus and caudate as a potential structural biomarker.

Identifying structural changes in brain networks that correlate with cognitive deficits in AD could lead to new diagnostic or therapeutic targets.

**Summary Of The Paper:**

The paper investigates the changes in white matter brain network communities in Alzheimer's disease (AD) using diffusion tensor imaging (DTI) data from the ADNI-GO/2 cohort. The study focuses on the modularity of structural brain networks, particularly examining the community structure of brain regions of interest (ROIs) across healthy control (HC), mild cognitive impairment (MCI), and AD groups. The authors employed consensus modularity optimization across a range of resolution parameters to identify community structures and observed that the thalamus and caudate become more segregated from the reward-based decision-making module in the AD group, forming an additional community not present in MCI or HC subpopulations. This segregation may be linked to cognitive deficits observed in AD, supporting the hypothesis that AD is a disconnection syndrome.

**Weaknesses:**

The AD group is relatively small (n=29), which may limit the generalizability of the findings. Larger sample sizes would provide more robust evidence.

The study acknowledges that network analysis is sensitive to the ROI parcellation scheme and other methodological choices, which could affect the reproducibility of the findings.

The study is cross-sectional, limiting its ability to track the progression of these network changes over time. Longitudinal studies could provide deeper insights into the evolution of network segregation in AD.

---

### Decision · Program_Chairs · 2024-09-23

Accept